# LARGE-SCALE REPRESENTATION LEARNING ON GRAPHS VIA BOOTSTRAPPING

**Shantanu Thakoor**[*]
DeepMind

**Corentin Tallec**
DeepMind

**Mohammad Gheshlaghi Azar**
DeepMind

**Mehdi Azabou**
Georgia Institute of Technology

**Eva Dyer**
Georgia Institute of Technology

**Rémi Munos**
DeepMind

**Petar Veličković**
DeepMind

**Michal Valko**
DeepMind

## ABSTRACT

Self-supervised learning provides a promising path towards eliminating the need for costly label information in representation learning on graphs. However, to achieve state-of-the-art performance, methods often need large numbers of negative examples and rely on complex augmentations. This can be prohibitively expensive, especially for large graphs. To address these challenges, we introduce Bootstrapped Graph Latents (BGRL) - a graph representation learning method that learns by predicting alternative augmentations of the input. BGRL uses only simple augmentations and alleviates the need for contrasting with negative examples, and is thus *scalable* by design. BGRL outperforms or matches prior methods on several established benchmarks, while achieving a 2-10x reduction in memory costs. Furthermore, we show that BGRL can be scaled up to extremely large graphs with hundreds of millions of nodes in the semi-supervised regime - achieving state-of-the-art performance and improving over supervised baselines where representations are shaped only through label information. In particular, our solution centered on BGRL constituted one of the winning entries to the Open Graph Benchmark - Large Scale Challenge at *KDD Cup 2021*, on a graph orders of magnitudes larger than all previously available benchmarks, thus demonstrating the scalability and effectiveness of our approach.

## 1 INTRODUCTION

Graphs provide a powerful abstraction for complex datasets that arise in a variety of applications such as social networks, transportation networks, and biological sciences (Hamilton et al., 2017; Derrow-Pinion et al., 2021; Zitnik & Leskovec, 2017; Chanussot et al., 2021). Despite recent advances in graph neural networks (GNNs), when trained with supervised data alone, these networks can easily overfit and may fail to generalize (Rong et al., 2019). Thus, finding ways to form simplified representations of graph-structured data without labels is an important yet unsolved challenge.

Current state-of-the-art methods for unsupervised representation learning on graphs (Veličković et al., 2019; Peng et al., 2020; Hassani & Khasahmadi, 2020; Zhu et al., 2020b;a; You et al., 2020) are *contrastive*. These methods work by pulling together representations of related objects and pushing apart representations of unrelated ones. For example, current best methods Zhu et al. (2020b) and Zhu et al. (2020a) learn node representations by creating two augmented versions of a graph, pulling together the representation of the same node in the two graphs, while pushing apart *every other node pair*. As such, they inherently rely on the ability to compare each object to a large number of *negatives*. In the absence of a principled way of choosing these negatives, this can require computation and memory quadratic in the number of nodes. In many cases, the generation of a large number of negatives poses a prohibitive cost, especially for large graphs.

---

[*]Correspondence to: Shantanu Thakoor <thakoor@google.com>.

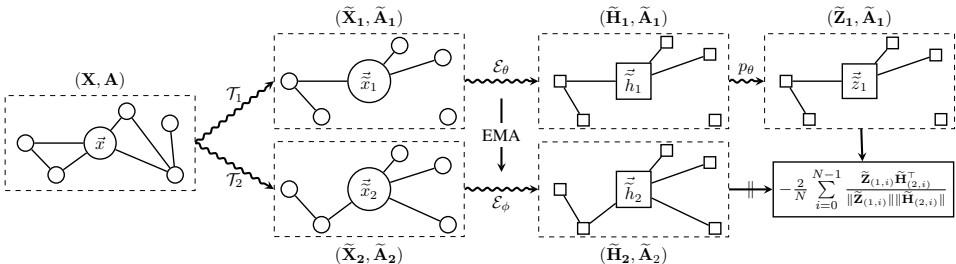

Figure 1: Overview of our proposed BGRL method. The original graph is first used to derive two different semantically similar views using augmentations $\mathcal{T}_{1,2}$. From these, we use encoders $\mathcal{E}_{\theta,\phi}$ to form online and target node embeddings. The predictor $p_\theta$ uses the online embedding $\widetilde{\mathbf{H}}_1$ to form a prediction $\widetilde{\mathbf{Z}}_1$ of the target embedding $\widetilde{\mathbf{H}}_2$. The final objective is then computed as the cosine similarity between $\widetilde{\mathbf{Z}}_1$ and $\widetilde{\mathbf{H}}_2$, flowing gradients only through $\widetilde{\mathbf{Z}}_1$. The target parameters $\phi$ are updated as an exponentially moving average of $\theta$.

In this paper, we introduce a scalable approach for self-supervised representation learning on graphs called *Bootstrapped Graph Latents* (BGRL). Inspired by recent advances in self-supervised learning in vision (Grill et al., 2020) , BGRL learns node representations by encoding two augmented versions of a graph using two distinct graph encoders: an online encoder, and a target encoder. The online encoder is trained through predicting the representation of the target encoder, while the target encoder is updated as an exponential moving average of the online network. Critically, BGRL does not require contrasting negative examples, and thus can scale easily to very large graphs.

Our main contributions are:

- We introduce Bootstrapped Graph Latents (BGRL), a graph self-supervised learning method that effectively scales to extremely large graphs and outperforms existing methods, while using only simple graph augmentations and not requiring negative examples (Section 2).

- We show that contrastive methods face a trade-off between peak performance and memory constraints, due to their reliance on negative examples (Section 4.2). Due to having time and space complexity scaling only *linearly* in the size of the input, BGRL avoids the performance-memory trade-off inherent to contrastive methods altogether. BGRL provides performance competitive with the best contrastive methods, while using 2-10x less memory on standard benchmarks (Section 3).

- We show that leveraging the scalability of BGRL allows making full use of the *vast amounts of unlabeled data* present in large graphs via semi-supervised learning. In particular, we find that efficient use of unlabeled data for representation learning prevents representations from overfitting to the classification task, and achieves significantly higher, state-of-the-art performance. This was critical to the success of our solution at *KDD Cup 2021* in which our BGRL-based solution was awarded one of the *winners*, on the largest publicly available graph dataset, of size 360GB consisting of 240 million nodes and 1.7 billion edges (Section 4.3).

## 2 BOOTSTRAPPED GRAPH LATENTS

### 2.1 BGRL COMPONENTS

BGRL builds representations through the use of two graph encoders, an online encoder $\mathcal{E}_\theta$ and a target encoder $\mathcal{E}_\phi$, where $\theta$ and $\phi$ denote two distinct sets of parameters. We consider a graph $\mathbf{G} = (\mathbf{X}, \mathbf{A})$, with *node features* $\mathbf{X} \in \mathbb{R}^{N \times F}$ and *adjacency matrix* $\mathbf{A} \in \mathbb{R}^{N \times N}$. BGRL first produces two alternate views of $\mathbf{G}$: $\mathbf{G}_1 = (\widetilde{\mathbf{X}}_1, \widetilde{\mathbf{A}}_1)$ and $\mathbf{G}_2 = (\widetilde{\mathbf{X}}_2, \widetilde{\mathbf{A}}_2)$, by applying stochastic graph augmentation functions $\mathcal{T}_1$ and $\mathcal{T}_2$ respectively. The online encoder produces an online representation from the first augmented graph, $\widetilde{\mathbf{H}}_1 := \mathcal{E}_\theta(\widetilde{\mathbf{X}}_1, \widetilde{\mathbf{A}}_1)$; similarly the target encoder produces a target representation of the second augmented graph, $\widetilde{\mathbf{H}}_2 := \mathcal{E}_\phi(\widetilde{\mathbf{X}}_2, \widetilde{\mathbf{A}}_2)$. The online representation is fed into a node-level predictor $p_\theta$ that outputs a prediction of the target representation, $\widetilde{\mathbf{Z}}_1 := p_\theta(\widetilde{\mathbf{H}}_1)$.

BGRL differs from prior bootstrapping approaches such as BYOL (Grill et al., 2020) in that it *does not use a projector network*. Unlike vision tasks, in which a projection step is used by BYOL for

dimensionality reduction, common embedding sizes are quite small for graph tasks and so this is not a concern in our case. In fact, we observe that this step can be eliminated altogether without loss in performance (Appendix B).

The augmentation functions $\mathcal{T}_1$ and $\mathcal{T}_2$ used are simple, standard graph perturbations previously explored (You et al., 2020; Zhu et al., 2020b). We use a combination of random **node feature masking** and **edge masking** with fixed masking probabilities $p_f$ and $p_e$ respectively. More details and background on graph augmentations is provided in Appendix D.

## 2.2 BGRL UPDATE STEP

**Updating the online encoder $\mathcal{E}_\theta$:** The online parameters $\theta$ (and not $\phi$), are updated to make the predicted target representations $\widetilde{\mathbf{Z}}_1$ closer to the true target representations $\widetilde{\mathbf{H}}_2$ for each node, by following the gradient of the cosine similarity w.r.t. $\theta$, i.e.,

$$\ell(\theta, \phi) = -\frac{2}{N} \sum_{i=0}^{N-1} \frac{\widetilde{\mathbf{Z}}_{(1,i)} \widetilde{\mathbf{H}}_{(2,i)}^\top}{\|\widetilde{\mathbf{Z}}_{(1,i)}\| \|\widetilde{\mathbf{H}}_{(2,i)}\|} \tag{1}$$

$$\theta \leftarrow \text{optimize}(\theta, \ \eta, \ \partial_\theta \ell(\theta, \phi)), \tag{2}$$

where $\eta$ is the learning rate and the final updates are computed from the gradients of the objective with respect to $\theta$ *only*, using an optimization method such as SGD or Adam (Kingma & Ba, 2015). In practice, we symmetrize this loss, by also predicting the target representation of the first view using the online representation of the second.

**Updating the target encoder $\mathcal{E}_\phi$:** The target parameters $\phi$ are updated as an exponential moving average of the online parameters $\theta$, using a decay rate $\tau$, i.e.,

$$\phi \leftarrow \tau\phi + (1 - \tau)\theta, \tag{3}$$

Figure 1 visually summarizes BGRL's architecture.

Note that although the objective $\ell(\theta, \phi)$ has undesirable or trivial solutions, BGRL does not actually optimize this loss. Only the online parameters $\theta$ are updated to reduce this loss, while the target parameters $\phi$ follow a different objective. This non-collapsing behavior even without relying on negatives has been studied further (Tian et al., 2021). We provide an empirical analysis of this behavior in Appendix A, showing that in practice BGRL does not collapse to trivial solutions and $\ell(\theta, \phi)$ does not converge to 0.

**Scalable non-contrastive objective:** Here we note that a contrastive approach would instead encourage $\widetilde{\mathbf{Z}}_{(1,i)}$ and $\widetilde{\mathbf{H}}_{(2,j)}$ to be far apart for node pairs $(i, j)$ that are dissimilar. In the absence of a principled way of choosing such dissimilar pairs, the naïve approach of simply contrasting *all pairs* $\{(i, j) \mid i \neq j\}$, scales *quadratically* in the size of the input. As BGRL does not rely on this contrastive step, BGRL scales *linearly* in the size of the graph, and thus is scalable by design.

## 3 COMPUTATIONAL COMPLEXITY ANALYSIS

We provide a brief description of the time and space complexities of the BGRL update step, and illustrate its advantages compared to previous strong contrastive methods such as GRACE (Zhu et al., 2020b), which perform a quadratic all-pairs contrastive computation at each update step. The same analysis applies to variations of the GRACE method such as GCA (Zhu et al., 2020a).

Consider a graph with $N$ nodes and $M$ edges, and simple encoders $\mathcal{E}$ that compute embeddings in time and space $\mathcal{O}(N + M)$. This property is satisfied by most popular GNN architectures such as convolutional (Kipf & Welling, 2017), attentional (Veličković et al., 2018), or message-passing (Gilmer et al., 2017) networks. BGRL performs four encoder computations per update step (twice for the target and online encoders, and twice for each augmentation) plus a node-level prediction step; GRACE performs two encoder computations (once for each augmentation), plus a node-level projection step. Both methods backpropagate the learning signal twice (once for each augmentation),

and we assume the backward pass to be approximately as costly as a forward pass. We ignore the cost for computation of the augmentations in this analysis. Thus the total time and space complexity per update step for `BGRL` is $6C_{\text{encoder}}(M + N) + 4C_{\text{prediction}}N + C_{\text{BGRL}}N$, compared to $4C_{\text{encoder}}(M + N) + 4C_{\text{projection}}N + C_{\text{GRACE}}N^2$ for `GRACE`, where $C.$ are constants depending on architecture of the different components. Table 1 shows an empirical comparison of `BGRL` and `GRACE`'s computational requirements on a set of benchmark tasks, with further details in Appendix I.

| Dataset | Amazon Photos | WikiCS | Amazon Computers | Coauthor CS | Coauthor Phy |
|---|---|---|---|---|---|
| #Nodes | 7,650 | 11,701 | 13,752 | 18,333 | 34,493 |
| #Edges | 119,081 | 216,123 | 245,861 | 81,894 | 247,962 |
| BGRL Memory | **0.47 GB** | **0.63 GB** | **0.58 GB** | **2.86 GB** | **5.50 GB** |
| GRACE Memory | 1.81 GB | 3.82 GB | 5.14 GB | 11.78 GB | OOM |

Table 1: Comparison of computational requirements on a set of standard benchmark graphs. OOM indicates ruuning out of memory on a 16GB V100 GPU.

## 4 Experimental Analysis

We present an extensive empirical study of performance and scalability, showing that `BGRL` is effective across a wide range of settings from frozen linear evaluation to semi-supervised learning, and both when performing full-graph training and training on subsampled node neighborhoods. We give results across a range of dataset scales and encoder architectures including convolutional, attentional, and message-passing neural networks.

We analyze the performance of `BGRL` on a set of 7 standard transductive and inductive benchmark tasks, as well as in the very high-data regime by evaluating on the MAG240M dataset (Hu et al., 2021). We present results on medium-sized datasets where contrastive objectives can be computed on the entire graph (Section 4.1), on larger datasets where this objective must be approximated (Section 4.2), and finally on the much larger MAG240M dataset designed to test scalability limits (Section 4.3), showing that `BGRL` improves performance across all scales of datasets. In Appendix C, we show that `BGRL` achieves state-of-the-art performance even in the low-data regime on a set of 4 small-scale datasets. Dataset sizes are summarized in Table 2 and described further in Appendix E.

**Evaluation protocol:** In most tasks, we follow the standard linear-evaluation protocol on graphs (Veličković et al., 2019). This involves first training each graph encoder in a fully unsupervised manner and computing embeddings for each node; a simple linear model is then trained on top of these frozen embeddings through a logistic regression loss with $\ell_2$ regularization, without flowing any gradients back to the graph encoder network. In the more challenging MAG240M task, we extend `BGRL` to the semi-supervised setting by combining our self-supervised representation learning loss together with a supervised loss. We show that `BGRL`'s bootstrapping objective obtains state-of-the-art performance in this hybrid setting, and even improves further with the added use of unlabeled data for representation learning - properties which have not been previously demonstrated by prior works on self-supervised representation learning on graphs.

Implementation details including model architectures and hyperparameters are provided in Appendix F. Algorithm implementation and experiment code for most tasks can be found at https://github.com/nerdslab/bgrl while code for our solution on MAG240M has been open-sourced as part of the *KDD Cup 2021* (Addanki et al., 2021) at https://github.com/deepmind/deepmind-research/tree/master/ogb_lsc/mag.

### 4.1 Performance and Efficiency gains when scalability is not a bottleneck

We first evaluate our method on a set of 5 recent real-world datasets — WikiCS, Amazon-Computers, Amazon-Photos, Coauthor-CS, Coauthor-Physics — in the transductive setting. Note that these are challenging medium-scale datasets specifically proposed for rigorous evaluation of semi-supervised node classification methods (Mernyei & Cangea, 2020; Shchur et al., 2018), but are almost all small enough that constrastive approaches such as `GRACE` (Zhu et al., 2020b) can compute their quadratic objective exactly. Thus, these experiments present a comparison of `BGRL` with prior methods in the

| | Task | Nodes | Edges | Features | Classes |
|---|---|---|---|---|---|
| **WikiCS** | Transductive | 11,701 | 216,123 | 300 | 10 |
| **Amazon Computers** | Transductive | 13,752 | 245,861 | 767 | 10 |
| **Amazon Photos** | Transductive | 7,650 | 119,081 | 745 | 8 |
| **Coauthor CS** | Transductive | 18,333 | 81,894 | 6,805 | 15 |
| **Coauthor Physics** | Transductive | 34,493 | 247,962 | 8,415 | 5 |
| **ogbn-arxiv** | Transductive | 169,343 | 1,166,243 | 128 | 40 |
| **PPI (24 graphs)** | Inductive | 56,944 | 818,716 | 50 | 121 (multilabel) |
| **MAG240M** | Transductive | 244,160,499 | 1,728,364,232 | 768 | 153 |

Table 2: Statistics of datasets used in our experiments.

idealized case where scalability is not a bottleneck. We show that even in this steelmanned setting, our method outperforms or matches prior methods while requiring a fraction of the memory costs.

We primarily compare `BGRL` against `GRACE`, a recent strong contrastive representation learning method on graphs. We also report performances for other commonly used self-supervised graph methods from previously published results (Perozzi et al., 2014; Veličković et al., 2019; Peng et al., 2020; Hassani & Khasahmadi, 2020; Zhu et al., 2020a), as well as `Random-Init` (Veličković et al., 2019), a baseline using embeddings from a randomly initialized encoder, thus measuring the quality of the inductive biases present in the encoder model. We use a 2-layer GCN model (Kipf & Welling, 2017) as our graph encoder $\mathcal{E}$, and closely follow models, architectures, and graph-augmentation settings used in prior works (Zhu et al., 2020a; Veličković et al., 2019; Zhu et al., 2020b).

| | WikiCS | Am. Comp. | Am. Photos | Co.CS | Co.Phy |
|---|---|---|---|---|---|
| Raw features | $71.98 \pm 0.00$ | $73.81 \pm 0.00$ | $78.53 \pm 0.00$ | $90.37 \pm 0.00$ | $93.58 \pm 0.00$ |
| `DeepWalk` | $74.35 \pm 0.06$ | $85.68 \pm 0.06$ | $89.44 \pm 0.11$ | $84.61 \pm 0.22$ | $91.77 \pm 0.15$ |
| `DeepWalk` + feat. | $77.21 \pm 0.03$ | $86.28 \pm 0.07$ | $90.05 \pm 0.08$ | $87.70 \pm 0.04$ | $94.90 \pm 0.09$ |
| `DGI` | $75.35 \pm 0.14$ | $83.95 \pm 0.47$ | $91.61 \pm 0.22$ | $92.15 \pm 0.63$ | $94.51 \pm 0.52$ |
| `GMI` | $74.85 \pm 0.08$ | $82.21 \pm 0.31$ | $90.68 \pm 0.17$ | OOM | OOM |
| `MVGRL` | $77.52 \pm 0.08$ | $87.52 \pm 0.11$ | $91.74 \pm 0.07$ | $92.11 \pm 0.12$ | $95.33 \pm 0.03$ |
| `Random-Init`$^\star$ | $78.95 \pm 0.58$ | $86.46 \pm 0.38$ | $92.08 \pm 0.48$ | $91.64 \pm 0.29$ | $93.71 \pm 0.29$ |
| `GRACE` $^\star$ | $\mathbf{80.14 \pm 0.48}$ | $89.53 \pm 0.35$ | $92.78 \pm 0.45$ | $91.12 \pm 0.20$ | OOM |
| `BGRL`$^\star$ | $79.98 \pm 0.10$ | $\mathbf{90.34 \pm 0.19}$ | $\mathbf{93.17 \pm 0.30}$ | $\mathbf{93.31 \pm 0.13}$ | $\mathbf{95.73 \pm 0.05}$ |
| GCA | $78.35 \pm 0.05$ | $88.94 \pm 0.15$ | $92.53 \pm 0.16$ | $93.10 \pm 0.01$ | $95.73 \pm 0.03$ |
| Supervised GCN | $77.19 \pm 0.12$ | $86.51 \pm 0.54$ | $92.42 \pm 0.22$ | $93.03 \pm 0.31$ | $95.65 \pm 0.16$ |

Table 3: Performance measured in terms of classification accuracy along with standard deviations. Our experiments, marked as $\star$, are over 20 random dataset splits and model initializations. The other results are taken from previously published reports. OOM indicates running out of memory on a 16GB V100 GPU. We report the best result for GCA out of the proposed GCA-DE, GCA-PR, and GCA-EV models.

In Table 3, we report results of our experiments on these standard benchmark tasks. We see that even when scalability does not prevent the use of contrastive objectives, `BGRL` performs competitively both with our unsupervised and fully supervised baselines, achieving state-of-the-art performance in 4 of the 5 datasets. Further, as noted in Table 1, `BGRL` achieves this despite using 2-10x less memory. `BGRL` provides this improvement in memory-efficiency at no cost in performance, demonstrating a useful practical advantage over prior methods such as `GRACE`.

**Effect of more complex augmentations:** In addition to the original `GRACE` method, we also highlight `GCA`, a variant of it that has the same learning objective but trades off more expressive but expensive graph augmentations for better performance. However, these augmentations often take time *cubic* in the size of the graph, or are otherwise cumbersome to implement on large graphs. As we focus on scalability to the high-data regime, we primarily restrict our comparisons to the base method `GRACE`, which uses the same simple, easily scalable augmentations as `BGRL`. Nevertheless, for the sake of completeness, in Table 4 we investigate the effect of these complex augmentations with `BGRL`. We see that `BGRL` obtains equivalent performance with both simple and complex augmentations, while `GCA` requires more expensive augmentations for peak performance. This indicates that `BGRL` can safely rely on simple augmentations when scaling to larger graphs without sacrificing performance.

| Method | Augmentation | Co.CS | Co.Phy | Am. Comp. | Am. Photos |
|--------|--------------|-------|--------|-----------|------------|
| BGRL | Standard | $93.31 \pm 0.13$ | $95.73 \pm 0.05$ | $90.34 \pm 0.19$ | $93.17 \pm 0.30$ |
| | Degree centrality | $93.34 \pm 0.13$ | $95.62 \pm 0.09$ | $90.39 \pm 0.22$ | $93.15 \pm 0.37$ |
| | Pagerank centrality | $93.34 \pm 0.11$ | $95.59 \pm 0.09$ | $90.45 \pm 0.25$ | $93.13 \pm 0.34$ |
| | Eigenvector centrality | $93.32 \pm 0.15$ | $95.62 \pm 0.06$ | $90.20 \pm 0.27$ | $93.03 \pm 0.39$ |
| GCA | Standard | $92.93 \pm 0.01$ | $95.26 \pm 0.02$ | $86.25 \pm 0.25$ | $92.15 \pm 0.24$ |
| | Degree centrality | $93.10 \pm 0.01$ | $95.68 \pm 0.05$ | $87.85 \pm 0.31$ | $92.49 \pm 0.09$ |
| | Pagerank centrality | $93.06 \pm 0.03$ | $95.72 \pm 0.03$ | $87.80 \pm 0.23$ | $92.53 \pm 0.16$ |
| | Eigenvector centrality | $92.95 \pm 0.13$ | $95.73 \pm 0.03$ | $87.54 \pm 0.49$ | $92.24 \pm 0.21$ |

Table 4: Comparison of BGRL and GCA for simple versus complex augmentation heuristics on four benchmark graphs. For GCA, we report the numbers provided in their original paper.

| | Validation | Test |
|---|-----------|------|
| MLP | $57.65 \pm 0.12$ | $55.50 \pm 0.23$ |
| node2vec | $71.29 \pm 0.13$ | $70.07 \pm 0.13$ |
| Random-Init$^\star$ | $69.90 \pm 0.11$ | $68.94 \pm 0.15$ |
| DGI$^\star$ | $71.26 \pm 0.11$ | $70.34 \pm 0.16$ |
| GRACE full-graph$^\star$ | OOM | OOM |
| GRACE-SUBSAMPLING $(k = 2)^\star$ | $60.49 \pm 3.72$ | $60.24 \pm 4.06$ |
| GRACE-SUBSAMPLING $(k = 8)^\star$ | $71.30 \pm 0.17$ | $70.33 \pm 0.18$ |
| GRACE-SUBSAMPLING $(k = 32)^\star$ | $72.18 \pm 0.16$ | $71.18 \pm 0.16$ |
| GRACE-SUBSAMPLING $(k = 2048)^\star$ | $\mathbf{72.61 \pm 0.15}$ | $\mathbf{71.51 \pm 0.11}$ |
| BGRL$^\star$ | $\mathbf{72.53 \pm 0.09}$ | $\mathbf{71.64 \pm 0.12}$ |
| Supervised GCN | $73.00 \pm 0.17$ | $71.74 \pm 0.29$ |

Table 5: Performance on the ogbn-arXiv task measured in terms of classification accuracy along with standard deviations. Our experiments, marked as $\star$, are averaged over 20 random model initializations. Other results are taken from previously published reports. OOM indicates running out of memory on a 16GB V100 GPU.

## 4.2 SCALABILITY-PERFORMANCE TRADE-OFFS FOR LARGE GRAPHS

When scaling up to large graphs, it may not be possible to compare each node's representation to all others. In this case, a natural way to reduce memory is to compare each node with only a subset of nodes in the rest of the graph. To study how the number of negatives impacts performance in this case, we propose an approximation of GRACE's objective called GRACE-SUBSAMPLING, where instead of contrasting every pair of nodes in the graph, we subsample $k$ nodes randomly across the graph to use as negative examples for each node at every gradient step. Note that $k = 2$ is the asymptotic equivalent of BGRL in terms of memory costs, as BGRL always only compares each node with itself across both views, i.e., BGRL faces no such computational difficulty or design choice in scaling up.

### EVALUATING ON OGBN-ARXIV DATASET

To study the tradeoff between performance and complexity we consider a node classification task on a much larger dataset, from the OGB benchmark (Hu et al., 2020a), ogbn-arXiv. In this case, GRACE cannot run without subsampling (on a GPU with 16GB of memory). Considering the increased difficulty of this task, we slightly expand our model to use 3 GCN layers, following the baseline model provided by Hu et al. (2020a). As there has not been prior work on applying GNN-based unsupervised approaches to the ogbn-arXiv task, we implement and compare against two representative contrastive-learning approaches, DGI and GRACE. In addition, we report results from Hu et al. (2020a) for node2vec (Grover & Leskovec, 2016) and a supervised-learning baseline. We report results on both validation and test sets, as is convention for this task since the dataset is split based on a chronological ordering.

Our results, summarized in Table 5, show that BGRL is competitive with the supervised learning baseline. Further, we note that the performance of GRACE-SUBSAMPLING is very sensitive to the parameter $k$—requiring a large number of negatives to match the performance of BGRL. Note that BGRL far exceeds the performance of GRACE-SUBSAMPLING with $k = 2$, its asymptotic equivalent in terms of memory; and that larger values of $k$ lead to out-of-memory errors on a 16GB GPU. These results suggest that the performance of contrastive methods such as GRACE may suffer due to approximations to their objectives that must be made when scaling up.

|  | PPI |
|---|---|
| Raw features | 42.20 |
| DGI | $63.80 \pm 0.20$ |
| GMI | $65.00 \pm 0.02$ |
| Random-Init | $62.60 \pm 0.20$ |
| GRACE MeanPooling encoder$^\star$ | $69.66 \pm 0.15$ |
| BGRL MeanPooling encoder$^\star$ | $69.41 \pm 0.15$ |
| GRACE GAT encoder$^\star$ | $69.71 \pm 0.17$ |
| BGRL GAT encoder$^\star$ | $\mathbf{70.49 \pm 0.05}$ |
| Supervised MeanPooling | *96.90 ± 0.20* |
| Supervised GAT | *97.30 ± 0.20* |

Table 6: Performance on the PPI task measured in terms of Micro-$F_1$ across the 121 labels along with standard deviations. Our experiments, marked as $\star$, are averaged over 20 random model initializations. Other results are taken from previously published reports.

EVALUATING ON PROTEIN-PROTEIN INTERACTION DATASET

Next, we consider the Protein-Protein Interaction (PPI) task—a more challenging inductive task on *multiple graphs* where the gap between the best self-supervised methods and fully supervised methods is (still) significantly large, due to 40% of the nodes missing feature information. In addition to simple mean-pooling propagation rules from GraphSage-GCN (Hamilton et al., 2017), we also consider Graph Attention Networks (GAT, Veličković et al., 2018) where each node aggregates features from its neighbors non-uniformly using a learned attention weight. It has been shown (Veličković et al., 2018) that GAT improves over non-attentional models on this dataset when trained in supervised settings, but these models have thus far not been able to be trained to a higher performance than non-attentional models through contrastive techniques.

We report our results in Table 6, showing that BGRL is competitive with GRACE when using the simpler MeanPooling networks. Applying BGRL to a GAT model, results in a new state-of-the-art performance, improving over the MeanPooling network. On the other hand, the GRACE contrastive loss is unable to improve performance of a GAT model over the non-attentional MeanPooling encoder. We observe that the approximation of the contrastive objective results not only in lower accuracies

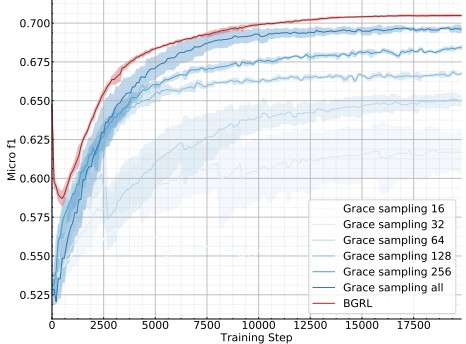

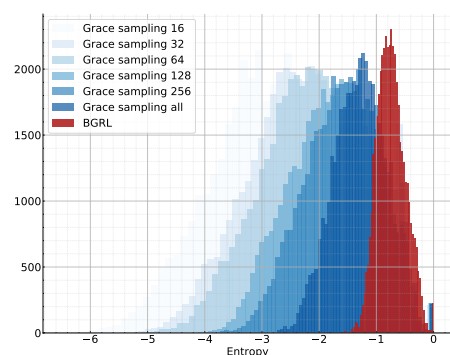

Figure 2: PPI task performance, averaged over 20 seeds.    Figure 3: Histogram of GAT attention entropies.

(Figure 2), but also qualitatively different behaviors in the GAT models being trained. In Figure 3, we examine the internals of GAT models trained through both BGRL and GRACE by analyzing the entropy of the attention weights learned. For each training node, we compute the average entropy of its attention weights across all GAT layers and attention heads, minus the entropy of a uniform attention distribution as a baseline. We see that GAT models learned using GRACE, particularly when subsampling few negative examples, tend to have very low attention entropies and perform poorly. On the other hand, BGRL is able to train the model to have meaningful attention weights, striking a balance between the low-entropy models learned through GRACE and the maximum-entropy

uniform-attention distribution. This aligns with recent observations (Kim & Oh, 2021; Wang et al., 2019) that auxiliary losses must be chosen carefully for the stability of GAT attention weights.

### 4.3 SCALING TO EXTREMELY LARGE GRAPHS

To further test the scalability and evaluate the performance of BGRL in the very high-data regime, we consider the MAG240M node classification task (Hu et al., 2021). As a single connected graph of 360GB with over 240 million nodes (of which 1.4 million are labeled) and 1.7 billion edges, this dataset is orders of magnitude larger than previously available public datasets, and poses a significant scaling challenge. Since the test labels for this dataset are (still) hidden, we report performance based on validation accuracies in our experiments. Implementation and experiment details are in Appendix G.

To account for the increased scale and difficulty of the classification task on this dataset, we make a number of changes to our learning setup. First, since we can no longer perform full-graph training due to the sheer size of the graph, we thus adopt the Neighborhood Sampling strategy proposed by Hamilton et al. (2017) to sample a small number of central nodes at which our loss is to be applied, and sampling a fixed size neighborhood around them. Second, we use more expressive Message Passing Neural Networks (Gilmer et al., 2017) as our graph encoders. Finally, as we are interested in pushing performance on this competition dataset, we make use of the available labels for representation learning and shift from evaluating on top of a frozen representation, to semi-supervised training by combining both supervised and self-supervised signals at each update step. We emphasize that these are significant changes from the standard small-scale evaluation setup for graph representation learning methods studied previously, and more closely resemble real-world conditions in which these algorithms would be employed.

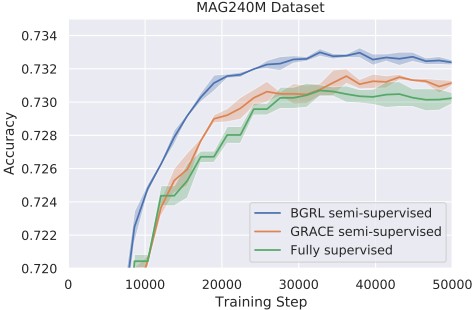

Figure 4: Performance on MAG240M using BGRL or GRACE-SUBSAMPLING as an auxiliary signal, averaged over 5 seeds and run for 50k steps.

Figure 5: Mixing varying amounts of unlabeled data for representation learning with BGRL, averaged over 5 seeds and run for 500k steps.

In Figure 4, we see that BGRL used as an auxiliary signal is able to learn faster and significantly improve final performance over fully supervised learning on this challenging task. Considering the difficulty of this task and the small gap in final performance between winning entries in the *KDD Cup 2021* contest, this is a significant improvement. On the other hand, GRACE-SUBSAMPLING provides much lower benefits over fully supervised learning, possibly due to no longer being able to sample sufficiently many negatives over the whole graph. Here we used $k = 256$ negatives, the largest value we were able to run without running out of memory.

We further show that we can leverage the high scalability of BGRL to make use of the vast amounts of *unlabeled* data present in the dataset. Since labeled nodes form only 0.5% of the graph, unlabeled data offers a rich self-supervised signal for learning better representations and ultimately improving performance on the supervised task. In Figure 5, we consider adding some number of unlabeled nodes to each minibatch of data, and examine the effect on performance as this ratio of unlabeled to labeled data in each batch increases. Thus at each step, we apply the supervised loss on only the labeled nodes in the batch, and BGRL on all nodes. Note that a ratio of 0 corresponds to the case where we apply BGRL as an auxiliary loss only to the training nodes, already examined in Figure 4. We observe a dramatic increase in both stability and peak performance as we increase this ratio, showing that BGRL can utilize the unlabeled nodes effectively to shape higher quality representations and prevent

early overfitting to the supervised signal. This effect shows steady improvement as we increase this ratio from 1x to 10x unlabeled data, where we stop due to resource constraints on running ablations on this large-scale graph - however, this trend may continue to even higher ratios, as the true ratio of unlabeled to labeled nodes present in the graph is 99x.

This result of 73.89% is **state-of-the-art** for this dataset for the highest single-model performance (i.e., without ensembling) - the OGB baselines report a score of 70.02% (Hu et al., 2021) while the *KDD Cup 2021* contest first place solution reported a score of 73.71% before ensembling.

**KDD Cup 2021:**[1]    Our solution using BGRL to shape representations, utilizing unlabeled data in conjunction with a supervised signal for semi-supervised learning, was awarded as one of the winners of the MAG240M track at OGB-LSC(Addanki et al., 2021). It achieved a final position of second overall, achieving 75.19% accuracy on the test set. The first and third place solutions achieved 75.49% and 74.60% respectively. Although differences in many other factors such as model architectures, feature engineering, ensembling strategies, etc. prevent a direct comparison[2] between these solutions, these results serve as a strong empirical evidence for the effectiveness of BGRL for learning representations on extremely large scale datasets.

## 5  RELATED WORK

Early methods in the area relied on *random-walk objectives* such as DeepWalk (Perozzi et al., 2014) and node2vec (Grover & Leskovec, 2016). Even though the graph neural networks (GNNs) inductive bias aligns with these objectives (Wu et al., 2019; Veličković et al., 2019; Kipf & Welling, 2017), composing GNNs and random-walks does not work very well and can even degrade performance (Hamilton et al., 2017). Earlier combinations of GNNs and self-supervised learning involve Embedding Propagation (García-Durán & Niepert, 2017), Variational Graph Autoencoders (Kipf & Welling, 2016) and Graph2Gauss (Bojchevski & Günnemann, 2018). Hu et al. (2020b) leverages BERT (Devlin et al., 2019) for representation learning in graph-structured inputs. Hu et al. (2020b) assumes specific graph structures and uses feature masking objectives to shape representations.

Recently, contrastive methods effective on images have also been adapted to graphs using GNNs. This includes DGI (Veličković et al., 2019), inspired by Deep InfoMax Hjelm et al. (2019), which contrasts node-local patches against global graph representations. Next, InfoGraph (Sun et al., 2020) modified DGI's pipeline for graph classification tasks. GMI Peng et al. (2020) maximizes a notion of *graphical* mutual information inspired by MINE (Belghazi et al., 2018), allowing for a more fine-grained contrastive loss than DGI's. The SimCLR method of Chen et al. (2020a;b) has been specialized for graphs by GRACE and variants such as GCA (Zhu et al., 2020b;a) that rely on more complex data-adaptive augmentations. GraphCL (You et al., 2020) adapts SimCLR to learn graph-level embeddings using a contrastive objective. MVGRL (Hassani & Khasahmadi, 2020) generalizes CMC (Tian et al., 2020) to graphs. Graph Barlow Twins (Bielak et al., 2021) presents a method to learn representations by minimizing correlation between different representation dimensions. Concurrent works DGB (Che et al., 2020) and SelfGNN (Kefato & Girdzijauskas, 2021), like BGRL, adapt BYOL (Grill et al., 2020) for graph representation learning. However, BGRL differs from these works in the following ways:

- We show that BGRL scales to and attains state-of-the-art results on the very high-data regime on the MAG240M dataset. These results are unprecedented in the graph self-supervised learning literature and demonstrate a high degree of scalability.

- We show that BGRL is effective even when trained on sampled subgraphs and not full graph.

- We provide an extensive analysis of the performance-computation trade-off of BGRL versus contrastive methods, showing that BGRL can be more efficient in terms of computation and memory usage as it requires no negative examples.

- We show that BGRL is effective when performing semi-supervised training, providing further gains when leveraging both labeled data and unlabeled data. This is a significant result that has not been demonstrated in graph representation learning methods using neural networks prior to our work.

---

[1]Leaderboard at https://ogb.stanford.edu/kddcup2021/results/#awardees_mag240m.

[2]For example, the first place solution used a much larger set of 30 ensembled models compared to our 10, and exclusively relied on architectural improvements to improve performance without using self-supervised learning.

## ICLR Ethics Statement

Our contributions are in developing and evaluating a general method for self-supervised representation learning in graphs. As such, they may be helpful in applications where obtaining labels can be challenging or expensive, thus enabling newer applications potentially in the direction of positive social good.

On the other hand, as an unsupervised pretraining method, there is a risk of practitioners using it for downstream tasks without carefully considering how these embeddings were originally trained, potentially leading to stereotyping or unfair biases. Further, since the bootstrapping dynamics of BGRL are not yet fully understood, there is a higher chance of it being used as a blackbox machine learning method and harmful downstream effects being difficult to diagnose and resolve.

## ICLR Reproducibility Statement

We believe that the results we report in this paper are reproducible and strengthen our empirical contributions.

We have submitted our algorithm implementation and experimental setup, config, and code for almost all of our experiments as supplementary material. Most experiments finish within 30 minutes on a single V100 GPU, and thus are easy to verify with few resources. In addition, we are providing trained model weights/checkpoints for directly loading and verifying performance without training. Experiments for which code has not been provided are described in detail in the appendices (Appendix E and Appendix F) and should allow for reproduction.

Besides this, code for our large-scale MAG240M solution has been open-sourced as part of the KDD Cup 2021 and has been verified independently by the OGB-LSC contest organizers.

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

## A  BGRL DOES NOT CONVERGE TO TRIVIAL SOLUTIONS

In Figure 6 we show the BGRL loss curve throughout training for all the datasets considered. As we see, the loss does not converge to zero, indicating that the training does not result in a trivial solution.

In Figure 7 we plot the spread of the node embeddings, i.e., the standard deviation of the representations learned across all nodes, divided by the average norm. As we see, the embeddings learned across all datasets have a standard deviation that is a similar order of magnitude as the norms of the embeddings themselves, further indicating that the training dynamics do not converge to a constant solution.

Further, Figure 8 shows that the embeddings do not collapse to zero or blow up as training progresses.

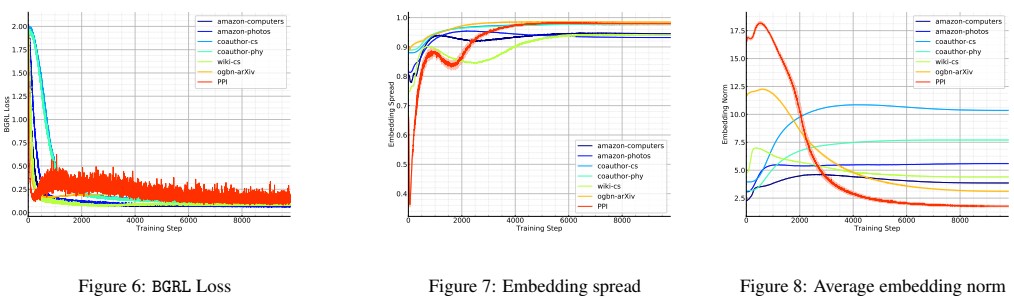

Figure 6: BGRL Loss          Figure 7: Embedding spread          Figure 8: Average embedding norm

## B  ABLATIONS ON PROJECTOR NETWORK

As noted in Section 2, BGRL does not use a projector network, unlike both BYOL and GRACE. Prior works such as GRACE use a projector network to prevent the embeddings from becoming completely invariant to the augmentations used - however in BGRL, the predictor network can serve the same purpose. On the other hand, BYOL relies on this for dimensionality reduction, to simplify the task of the predictor $p_\theta$, as it is challenging to directly predict very high-dimensional embeddings. required for large-scale vision tasks like ImageNet (Deng et al., 2009). Here we empirically verify that even in our most challenging, large-scale task of MAG240M, the projector network is not needed and only slows down learning. In Figure 9 we can see that adding the projector network leads to both slower learning and a lower final performance.

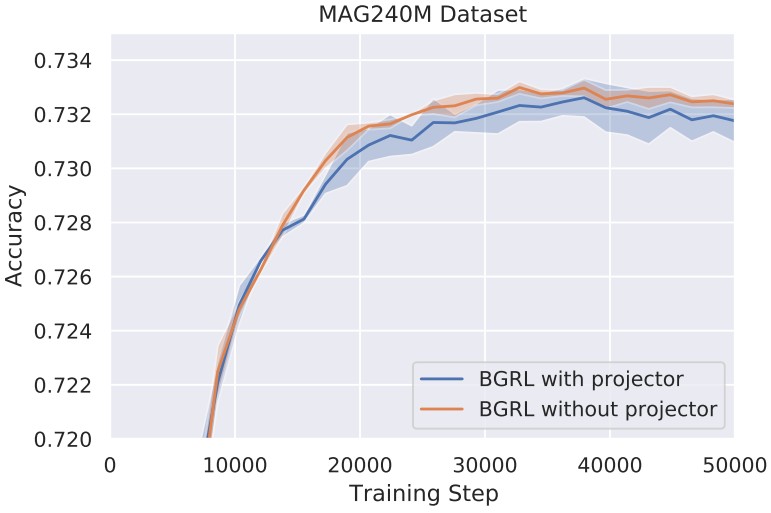

Figure 9: Performance on OGB-LSC MAG240M task, averaged over 5 seeds, testing effect of using projector network.

## C    COMPARISON ON SMALL DATASETS

We perform additional experiments on 4 commonly used small datasets, (Cora, CiteSeer, PubMed, and DBLP) (Sen et al., 2008; Bojchevski & Günnemann, 2017) and show that BGRL's bootstrapping mechanism still performs well in the low-data regime, even attaining a new state of the art performance on two of the datasets.

Note that Table 7 reports results averaged over 20 random dataset splits, as has been followed in Zhu et al. (2020b), instead of using the standard fixed splits for these datasets which are known to be easy to unreliable for evaluating GNN methods (Shchur et al., 2018).

|            | Cora          | CiteSeer      | PubMed        | DBLP          |
|------------|---------------|---------------|---------------|---------------|
| GRACE      | $83.02 \pm 0.89$ | $71.63 \pm 0.64$ | $86.06 \pm 0.26$ | $84.08 \pm 0.29$ |
| BGRL       | $83.83 \pm 1.61$ | $72.32 \pm 0.89$ | $86.03 \pm 0.33$ | $84.07 \pm 0.23$ |
| Supervised | 82.8          | 72.0          | 84.9          | 82.7          |

Table 7: Evaluation on small datasets. Results averaged over 20 dataset splits and model initializations.

## D    GRAPH AUGMENTATION FUNCTIONS

Generating meaningful augmentations is a much less explored problem in graphs than in other domains such as vision. Further, since we work over entire graphs, complex augmentations can be very expensive to compute and will impact all nodes at once. Our contributions are orthogonal to this problem, and we primarily consider only the standard graph augmentation pipeline that has been used in previous works on representation learning (You et al., 2020; Zhu et al., 2020b).

In particular, we consider two simple graph augmentation functions — **node feature masking** and **edge masking**. These augmentations are graph-wise: they do not operate on each node independently, and instead leverage graph topology information through edge masking. This contrasts with transformations used in BYOL, which operate on each image independently. First, we generate a single random binary mask of size $F$, each element of which follows a Bernoulli distribution $\mathcal{B}(1 - p_f)$, and use it to **mask features** of all nodes in the graph (i.e., all nodes have the same features masked). Empirically, we found that performance is similar for using different random masks per node or sharing them, and so we use a single mask for simplicity. In addition to this node-level attribute transformation, we also compute a binary mask of size $E$ (where $E$ is the number of edges in the original graph), each element of which follows a Bernoulli distribution $\mathcal{B}(1 - p_e)$, and use it to **mask edges** in the augmented graph. To compute our final augmented graphs, we make use of both augmentation functions with different hyperparameters for each graph, i.e. $p_{f_1}$ and $p_{e_1}$ for the first view, and $p_{f_2}$ and $p_{e_2}$ for the second view.

Beyond these standard augmentations, in Section 4.1 we also consider more complex *adaptive* augmentations proposed by prior works (Zhu et al., 2020a) which use various heuristics to mask different features or edges with different probabilities.

## E    DATASET DETAILS

**WikiCS**[3]    This graph is constructed from Wikipedia references, with nodes representing articles about Computer Science and edges representing links between them. Articles are classified into 10 classes based on their subfield, and node features are the average of GloVE (Pennington et al., 2014) embeddings of all words in the article. This dataset comes with 20 canonical train/valid/test splits, which we use directly.

**Amazon Computers, Amazon Photos**[4]    These graphs are from the Amazon co-purchase graph (McAuley et al., 2015) with nodes representing products and edges being between pairs of goods frequently purchased together. Products are classified into 10 (for Computers) and 8 (for Photos)

---

[3]https://github.com/pmernyei/wiki-cs-dataset/raw/master/dataset
[4]https://github.com/shchur/gnn-benchmark/tree/master/data/npz

classes based on product category, and node features are a bag-of-words representation of a product's reviews. We use a random split of the nodes into (10/10/80%) train/validation/test nodes respectively as these datasets do not come with a standard dataset split.

**Coauthor CS, Coauthor Physics**[5]    These graphs are from the Microsoft Academic Graph (Sinha et al., 2015), with nodes representing authors and edges between authors who have co-authored a paper. Authors are classified into 15 (for CS) and 5 (for Physics) classes based on the author's research field, and node features are a bag-of-words representation of the keywords of an author's papers. We again use a random (10/10/80%) split for these datasets.

**ogbn-arXiv:**    This is another citation network, where nodes represent CS papers on arXiv indexed by the Microsoft Academic Graph (Sinha et al., 2015). In our experiments, we symmetrize this graph and thus there is an edge between any pair of nodes if one paper has cited the other. Papers are classified into 40 classes based on arXiv subject area. The node features are computed as the average word-embedding of all words in the paper, where the embeddings are computed using a skip-gram model (Mikolov et al., 2013) over the entire corpus.

**PPI** [6]    is a protein-protein interaction network (Zitnik & Leskovec, 2017; Hamilton et al., 2017), comprised of multiple (24) graphs each corresponding to different human tissues. We use the standard dataset split as 20 graphs for training, 2 for validation, and 2 for testing. Each node has 50 features computed from various biological properties. This is a multilabel classification task, where each node can possess up to 121 labels.

## F    IMPLEMENTATION DETAILS

In all our experiments, we use the AdamW optimizer (Kingma & Ba, 2015; Gugger & Howard, 2018) with weight decay set to $10^{-5}$, and all models initialized using Glorot initialization (Glorot & Bengio, 2010). The BGRL predictor $p_\theta$ used to predict the embedding of nodes across views is fixed to be a Multilayer Perceptron (MLP) with a single hidden layer. The decay rate $\tau$ controlling the rate of updates of the BGRL target parameters $\phi$ is initialized to 0.99 and gradually increased to 1.0 over the course of training following a cosine schedule. Other model architecture and training details vary per dataset and are described further below. The augmentation hyperparameters $p_{f_{1,2}}$ and $p_{e_{1,2}}$ are reported below.

**Graph Convolutional Networks**    Formally, the GCN propagation rule (Kipf & Welling, 2017) for a single layer is as follows,

$$\text{GCN}_i(\mathbf{X}, \mathbf{A}) = \sigma\left(\widehat{\mathbf{D}}^{-\frac{1}{2}}\widehat{\mathbf{A}}\widehat{\mathbf{D}}^{-\frac{1}{2}}\mathbf{X}\mathbf{W}_i\right), \tag{4}$$

where $\widehat{\mathbf{A}} = \mathbf{A} + \mathbf{I}$ is the adjacency matrix with self-loops, $\widehat{\mathbf{D}}$ is the degree matrix, $\sigma$ is a non-linearity such as ReLU, and $\mathbf{W}_i$ is a learned weight matrix for the $i$'th layer.

**Mean Pooling Rule**    Formally, the Mean Pooling (Hamilton et al., 2017) rule for a single layer is given by:

$$\text{MP}_i(\mathbf{X}, \mathbf{A}) = \sigma(\widehat{\mathbf{D}}^{-1}\widehat{\mathbf{A}}\mathbf{X}\mathbf{W}_i) \tag{5}$$

As proposed by Veličković et al. (2019), our exact encoder in inductive experiments $\mathcal{E}$ is a 3-layer mean-pooling network with skip connections. We use a layer size of 512 and PReLU (He et al., 2015) activation. Thus, we compute:

$$\mathbf{H}_1 = \sigma(\text{MP}_1(\mathbf{X}, \mathbf{A})) \tag{6}$$

$$\mathbf{H}_2 = \sigma(\text{MP}_2(\mathbf{H}_1 + \mathbf{X}\mathbf{W}_{skip}, \mathbf{A})) \tag{7}$$

$$\mathcal{E}(\mathbf{X}, \mathbf{A}) = \sigma(\text{MP}_3(\mathbf{H}_2 + \mathbf{H}_1 + \mathbf{X}\mathbf{W}_{skip'}, \mathbf{A})) \tag{8}$$

---

[5] https://github.com/shchur/gnn-benchmark/tree/master/data/npz
[6] https://s3.us-east-2.amazonaws.com/dgl.ai/dataset/ppi.zip

**Graph Attention Networks**    The GAT layer (Veličković et al., 2018) consists of a learned matrix $\mathbf{W}$ that transforms each node features. We then use self-attention to compute attention coefficient for a pair of nodes $i$ and $j$ as $e_{ij} = a(\mathbf{h}_i, \mathbf{h}_j)$. The attention function $a$ is computed as LeakyReLU($\mathbf{a}[\mathbf{Wh}_i||\mathbf{Wh}_j]$), where $\mathbf{a}$ is a learned matrix transforming a pair of concatenated attention queries into a single scalar attention logit. The weight of the edge between nodes $i$ and $j$ is computed as $\alpha_{ij} = \text{softmax}_j(e_{ij})$. We follow the architecture proposed by Veličković et al. (2018), including a 3-layer GAT model (with the first 2 layers consisting of 4 heads of size 256 each and the final layer size 512 with 6 output heads), ELU activation (Clevert et al., 2016), and skip-connections in intermediate layers.

**Model architectures**    As described in Section 4, we use GCN (Kipf & Welling, 2017) encoders in our experiments on the smaller transductive tasks, while on the inductive task of PPI we use MeanPooling encoders with residual connections. The BGRL predictor $p_\theta$ is implemented as a mutilayer perceptron (MLP). We also used stabilization techniques like batch normalization (Ioffe & Szegedy, 2015), layer normalization (Ba et al., 2016), and weight standardization (Qiao et al., 2019). The decay rate use for statistics in the batch normalization is fixed to 0.99. We use PReLU activation (He et al., 2015) in all experiments except those using a GAT encoder, where we use the ELU activation (Clevert et al., 2016). In all our models, at each layer including the final layer, we apply first the batch/layer normalization as applicable, and then the activation function. Table 8 describes hyperparameter and architectural details for most of our experimental setups with BGRL. In addition to these standard settings, we perform additional experiments on the PPI dataset using a GAT (Veličković et al., 2018) model as the encoder. When using the GAT encoder on PPI, we use 3 attention layers — the first two with 4 attention heads of size 256 each, and the final with 6 attention heads of size 512, following a very similar model proposed by Veličković et al. (2018). We concatenate the attention head outputs for the first 2 layers, and use the mean for the final output. We also use the ELU activation (Clevert et al., 2016), and skip connections in the intermediate attention layers, as suggested by Veličković et al. (2018).

| Dataset | WikiCS | Am. Computers | Am. Photos | Co. CS | Co. Physics | ogbn-arXiv | PPI |
|---|---|---|---|---|---|---|---|
| $p_{f,1}$ | 0.2 | 0.2 | 0.1 | 0.3 | 0.1 | 0.0 | 0.25 |
| $p_{f,2}$ | 0.1 | 0.1 | 0.2 | 0.4 | 0.4 | 0.0 | 0.00 |
| $p_{e,1}$ | 0.2 | 0.5 | 0.4 | 0.3 | 0.4 | 0.6 | 0.30 |
| $p_{e,2}$ | 0.3 | 0.4 | 0.1 | 0.2 | 0.1 | 0.6 | 0.25 |
| $\eta_{\text{base}}$ | $5 \cdot 10^{-4}$ | $5 \cdot 10^{-4}$ | $10^{-4}$ | $10^{-5}$ | $10^{-5}$ | $10^{-2}$ | $5 \cdot 10^{-3}$ |
| embedding size | 256 | 128 | 256 | 256 | 128 | 256 | 512 |
| $\mathcal{E}$ hidden sizes | 512 | 256 | 512 | 512 | 256 | 256, 256 | 512, 512 |
| $p_\theta$ hidden sizes | 512 | 512 | 512 | 512 | 512 | 256 | 512 |
| batch norm | Y | Y | Y | Y | Y | N | N |
| layer norm | N | N | N | N | N | Y | Y |
| weight standard. | N | N | N | N | N | Y | N |

Table 8: Hyperparameter settings for unsupervised BGRL learning.

**Augmentation parameters**    The hyperparameter settings for graph augmentations, as well as the sizes of the embeddings and hidden layers, very closely follow previous work (Zhu et al., 2020b;a) on all datasets with the exception of ogbn-arXiv. On this dataset, since there has not been prior work on applying self-supervised graph learning methods, we provide the hyperparameters we found through a small grid search.

**Optimization settings**    We perform full-graph training at each gradient step on all small-scale experiments, with the exception of experiments using GAT encoders on the PPI dataset. Here, due to memory constraints, we perform training with a batch size of 1 graph. Since the PPI dataset consists of multiple smaller, disjoint subgraphs, we do not have to perform any node subsampling at training time.

We use Glorot initialization (Glorot & Bengio, 2010) the AdamW optimizer (Kingma & Ba, 2015; Gugger & Howard, 2018) with a base learning rate $\eta_{\text{base}}$ and weight decay set to $10^{-5}$. The learning rate is annealed using a cosine schedule over the course of learning of $n_{\text{total}}$ total steps with an initial warmup period of $n_{\text{warmup}}$ steps. Hence, the learning rate at step $i$ is computed as

$$\eta_i \triangleq \begin{cases} \frac{i \times \eta_{\text{base}}}{n_{\text{warmup}}} & \text{if } i \leq n_{\text{warmup}}, \\ \eta_{\text{base}} \times \left(1 + \cos \frac{(i - n_{\text{warmup}}) \times \pi}{n_{\text{total}} - n_{\text{warmup}}}\right) \times 0.5 & \text{if } n_{\text{warmup}} \leq i \leq n_{\text{total}}. \end{cases}$$

We fix $n_{\text{total}}$ to be 10,000 total steps and $n_{\text{warmup}}$ to 1,000 warmup steps, with the exception of experiments on the GAT encoder that requires using a batch size of 1 graph on the PPI dataset. In this case, we increase the number of total steps to 20,000 and warmup to 2,000 steps.

The target network parameters $\phi$ are initialized randomly from the same distribution of the online parameters $\theta$ but with a different random seed. The decay parameter $\tau$ is also updated using a cosine schedule starting from an initial value of $\tau_{\text{base}} = 0.99$ and is computed as

$$\tau_i \triangleq 1 - \frac{(1 - \tau_{\text{base}})}{2} \times \left(\cos\left(\frac{i \times \pi}{n_{\text{total}}}\right) + 1\right).$$

These annealing schedules for both $\eta$ and $\tau$ follow the procedure used by Grill et al. (2020).

**Frozen linear evaluation of embeddings**   In the linear evaluation protocol, the final evaluation is done by fitting a linear classifier on top of the frozen learned embeddings without flowing any gradients back to the encoder. For the smaller datasets of WikiCS, Amazon Computers/Photos, and Coauthor CS/Physics, we use an $\ell_2$-regularized LogisticRegression classifier from Scikit-Learn (Pedregosa et al., 2011) using the 'liblinear' solver. We do a hyperparameter search over the regularization strength to be between $\{2^{-10}, 2^{-9}, \ldots 2^9, 2^{10}\}$.

For larger PPI and ogbn-arXiv datasets, where the liblinear solver takes too long to converge, we instead perform 100 steps of gradient descent using AdamW with learning rate 0.01, with a smaller hyperparameter search on the weight decay between $\{2^{-10}, 2^{-8}, 2^{-6}, \ldots 2^6, 2^8, 2^{10}\}$.

In all cases, we $\ell_2$-normalize the frozen learned embeddings over the entire graph before fitting the classifier on top.

## G   MAG240M EXPERIMENT DETAILS

Full implementation and experiment code has been open-sourced as part of the KDD Cup 2021. Key implementation details and hyperparameter descriptions are reproduced below.

**OGB-LSC MAG240M Dataset:**   This is a heterogeneous graph introduced for the KDD Cup 2021 (Hu et al., 2021), comprised of 121 million academic papers, 122 million authors, and 26 thousand institutions. Papers are represented by 768-dimensional BERT embeddings (Devlin et al., 2019), and the task is to classify arXiv papers into one of 153 categories, where 1% of the paper nodes are from arXiv.

**Message Passing Neural Networks encoders:**   We use a bi-directional version of the standard MPNN (Gilmer et al., 2017) architectures with 4 message passing steps, a hidden size of 256 at each layer, with node and edge update functions represented by Multilayer Perceptrons (MLPs) with 2 hidden layers of size 512 each.

**Node Neighborhood Sampling:**   Since we can no longer perform full-graph training, we sample a batch size of 1024 central nodes split across 8 devices, and subsample a fixed-size neighborhood for each. Specifically, we sample a depth-2 neighborhood with different numbers of neighbors sampled per layer depending on the type (paper, author, institution) of each neighbor. We sample up to 80 papers and 20 authors for each paper; and 40 papers and 10 institutions per author.

**Other hyperparamters:**   We use edge masking probability $p_e$ of 0.2 and feature masking probability $p_f$ of 0.4 for each augmentation. We use a higher decay rate $\tau$, starting at 0.999 and decayed to 1.0 with a cosine schedule. We use AdamW optimizer with a weight decay of $10^{-5}$, and a learning rate starting at 0.01 and annealed to 0 over the course of learning with a warmup step equal to 10% the period of learning.

## H    MODEL ABLATIONS ON MAG240M EXPERIMENTS

In our main experiments on the OGB-LSC MAG240M dataset, we focus on MPNN encoders to *(i)* achieve high accuracy on this challenging dataset, and *(ii)* to evaluate the stability of training these more complex models with the `BGRL` approach. In this section, we further experiment with simpler GCN encoders and evaluate the benefits of applying `BGRL` even on top of these weaker encoders.

We use a 2-layer GCN encoder, with an embedding size of 128. All other settings such as learning rate schedules, augmentation parameters, etc. are unchanged.

In Figure 10, we see that both `BGRL` and `GRACE` improve over the performance of a fully supervised approach, with `BGRL` learning faster and more stably. Thus the effectiveness of `BGRL` even with weaker encoder architectures makes it more applicable in practice.

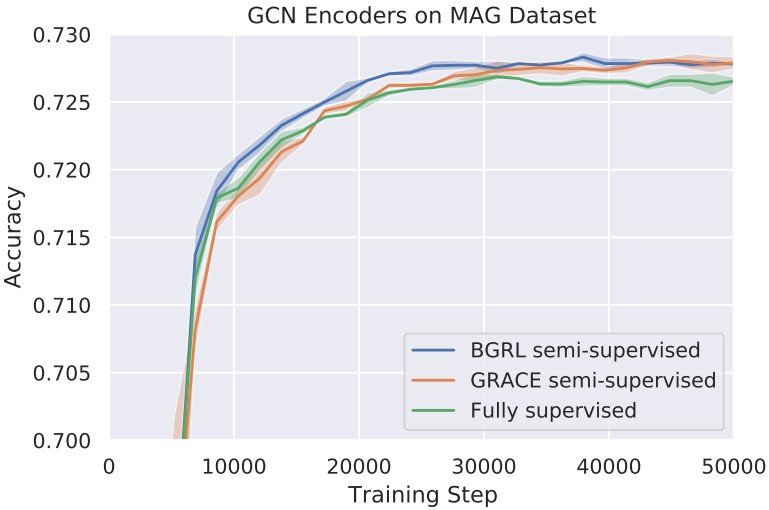

Figure 10: Performance on OGB-LSC MAG240M task, averaged over 3 seeds, using GCN encoders.

## I    VISUALIZATION OF SCALING BEHAVIOR

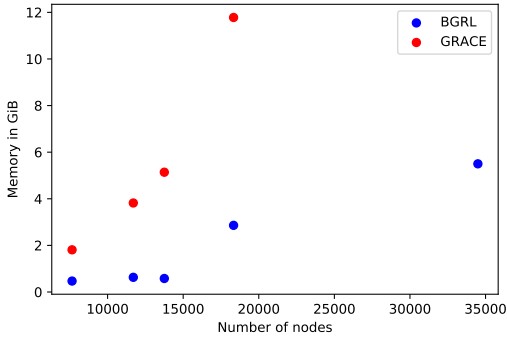

Figure 11: Memory usage of `BGRL` and `GRACE` across 5 standard datasets.

In this section, we provide the information contained in Table 1 as a scatterplot, to more easily visualize the different scaling properties of `BGRL` and `GRACE`. We see in Figure 11 that the empirical scaling behavior matches the theoretical predictions in Section 3. Note that we do not show the memory usage of `GRACE` for the largest dataset, as it runs out of memory here, and we only visualize as a function of the number of nodes in the graph and ignore the number of edges for the purposes of this visualization.

## J    FROZEN LINEAR EVALUATION ON MAG240M

We briefly run experiments evaluating `BGRL` and `GRACE` under the frozen evaluation protocol using an MLP classification layer on the MAG240M dataset. We see that although `GRACE` outperforms BGRL in this setting, both methods perform poorly. In particular, they underperform `LabelProp` (Zhu & Ghahramani, 2002; Hu et al., 2021) a simple parameterless, graph-agnostic baseline. This, combined with our goal of pushing performance on the competition dataset, motivates our consideration of the semi-supervised learning setting.

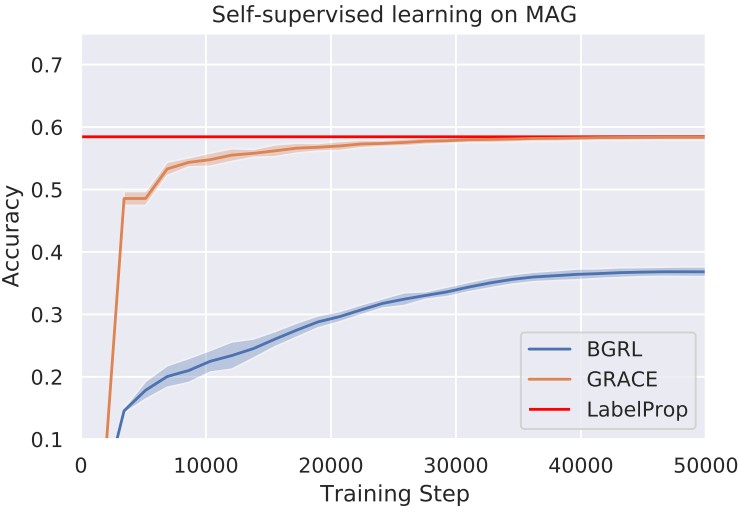

Figure 12: Performance on OGB-LSC MAG240M task, averaged over 5 seeds, under frozen evaluation protocol.

