# OpenReview forum: "Large-Scale Representation Learning on Graphs via Bootstrapping"
_ICLR.cc/2022/Conference — ICLR 2022 Poster_

### Official Review · Reviewer_8P5B · 2021-11-02

**Correctness:** 3
**Technical Novelty And Significance:** 2
**Empirical Novelty And Significance:** 4
**Recommendation:** 5
**Confidence:** 3

**Main Review:**

# Strengths

* Different from contrastive methods, the cost of computing the loss for BRGL exactly scales only linearly with graph size.
* An extensive suite of experiments shows BRGL achieves SOTA while avoiding the performance-memory tradeoff of contrastive methods.
* BRGL was validated in practice at the KDD cup 2021.

# Weaknesses
* BRGL is a direct analogue of BYOL for graphs (but without a projector network). Therefore, the novelty here is relatively limited.
* The authors provide no intuition on why their method works.

# Observations
*  It might help to show Table 1 as scatter plots nodes/edges versus memory. This way, we could ‘visualize’ the asymptotic difference between GRACE and BRGL.
* From the description in Section 4.2, GRACE-SUBSAMPLING scales linearly on $k$. If this is the case, $k=2$ asymptotically implies the same complexity as $k=2048$.

**Summary Of The Paper:**

This paper proposes BRGL, a method for graph representation learning based on ‘bootstrapping’ (in the same sense BYOL is). Different from the prior art, which mainly relies on contrastive learning, BRGL naturally scales linearly with graph size.  An extensive suite of experiments shows that BRGL scales better than previous works while yielding state-of-the-art (SOTA) performance in node classification.

**Summary Of The Review:**

The paper shows good results, but i) the novelty is relatively limited; and ii) there is no intuition on why the method works. Therefore, I believe the paper is just below borderline.

---

> ### Author Response · Authors · 2021-11-17
> **Response to Reviewer 8P5B**
>
> Thank you for your review and useful suggestions. We are happy that you found our experiments on the performance-memory tradeoff encountered in graph representation learning a strength of the paper.
>
> In regards to your points about the novelty, please see our general response for a more detailed discussion of the contributions of this paper. We would like to emphasize that this work provides a new study of the tradeoff between performance and memory in contrastive learning, and that we show how BGRL can provide a good balance between the two. We have not seen this type of comparison in the literature and we believe that this aspect of the work, combined with our strong empirical comparisons make the work worthy of publication. We hope that you will reconsider your score.
>
> (1) Intuition behind BGRL:  We thank the reviewer for their suggestion about including more intuition about how and why the method works. While we agree that the paper could benefit from some added intuitions, the focus of this paper is mainly on adapting BYOL to the graph setting and showing how the removal of negative examples via bootstrapping allows us to scale to much larger graphs than with contrastive methods. In a revised version, we will include a discussion of the intuition behind BYOL and recent work which investigates the necessity of different components in non-contrastive methods [1, 2].
>
> (2) Improved visualization of nodes/edges versus memory:  Thank you for this suggestion, we have included a scatterplot visualization of the information in Table 1 for better understanding of asymptotic behavior (see Appendix I, Figure 11).
>
>
> (3) Scaling of GRACE-SUBSAMPLING: We will update the text to make it clear that the k=2 case is the closest equivalent of BGRL in the sense of requiring approximately the same number of comparisons between node embeddings.
>
> [1] Y. Tian, X. Chen, and S. Ganguli, Understanding Self-Supervised Learning Dynamics without Contrastive Pairs, in ICML, 2021.
>
> [2] Chen, X. and He, K. Exploring simple siamese representation learning. arXiv preprint arXiv:2011.10566, 2020.

---

### Official Review · Reviewer_8wKC · 2021-11-05

**Correctness:** 3
**Technical Novelty And Significance:** 2
**Empirical Novelty And Significance:** 3
**Recommendation:** 6
**Confidence:** 5

**Main Review:**

The paper can be evaluated mainly from two perspectives:

1. The novelty of the method and the conceptual contribution is extremely limited since the proposed model is a pretty straightforward application of the BYOL [1] model on graph data.  The difference of not having a projection head cannot be deemed as a contribution or novelty, since the reason is the low dimension of the node features in the graph datasets, as mentioned by the authors.

2. The experimental contribution is very solid. The model is evaluated on multiple large datasets and is also tested via the KDD CUP competition and promising results are obtained. This would be a significant contribution to the community and would facilitate future researches in this direction.

Besides the two main points raised above, the method section could be further improved by including explanations on the motivation of the designs, which is especially important for readers who are not familiar with BYOL. Currently, the paper just described the operations of the model without explanation on the operations are designed in this way. Explanations from the following points may be considered: 1. What is the difference between the online and the target branch. 2. What makes the target branch become a target for the online branch? 3. Why is the target branch updated through EMA? 4. What objective does the target branch (\phi) follows, as mentioned in the paragraph under Eq(3), in page 3.

[1] Grill, Jean-Bastien, et al. "Bootstrap your own latent: A new approach to self-supervised learning." arXiv preprint arXiv:2006.07733 (2020).

**Summary Of The Paper:**

This paper introduces Bootstrapped Graph Latents (BGRL) to learn graph representation by predicting alternative augmentations of the input.  The experiment results on a wide range of datasets exhibit appealing performance.

**Summary Of The Review:**

Considering the empirical contribution of the paper, I am inclined to accept this paper.

---

> ### Author Response · Authors · 2021-11-17
> **Response to Reviewer 8wKC**
>
> Thank you for the useful suggestions and overall positive review. We are glad you found the “experimental contribution very solid” and “a significant contribution to the community”. We would like to point out that we have prepared an open-source release of the code which will further enable reproducibility of our results and thus help the community research in this direction.
>
> (1) In regards to your points about the novelty, please see our general response for a more detailed discussion of the contributions of this paper. We emphasise that this work provides a new study of the tradeoff between performance and memory in contrastive learning, and that BGRL can provide a good balance between the two without needing any complex augmentations. We have not seen this type of comparison in the literature and we believe that this aspect of the work, combined with our strong empirical comparisons make the work worthy of publication.
>
> (2) Improvements to methods section:  We thank the reviewer for pointing out some of the areas where we could improve our explanations of the method, especially for those readers unfamiliar with BYOL. In a revised version, we will add more intuition behind the method. In particular, we will aim to provide a better explanation of the update of the target network with an EMA and discuss alternative approaches explored in the literature.
>
> **Proposed revisions** (to be added in Section 2.2, after the description of “Updating the target encoder”):
>
> Note that although the objective $\ell(\theta, \phi)$ has undesirable or trivial solutions,
> BGRL does not actually optimize this loss for both the online and target networks. Only the online parameters $\theta$ are updated, while the target parameters $\phi$ are updated according to an exponential moving average (EMA). Thus the target branch can be viewed as a slowly moving version of the online network that is used to provide *stable targets* for our prediction task. Defining the target network as an EMA of the online network is a simple approach that works well in practice for improving stability, and has seen prior use in contrastive representation learning [2] and reinforcement learning [3]. More recent analyses [1] of non-contrastive methods [4, 5] have studied the impact of different components (the predictor network, EMA updates for the target network, stop-gradient) of this family of methods to understand their strong performance on complex tasks in vision.
>
> [1] Y. Tian, X. Chen, and S. Ganguli, Understanding Self-Supervised Learning Dynamics without Contrastive Pairs, in ICML, 2021.
>
> [2] Kaiming He et al,  Momentum contrast for unsupervised visual representation learning. arXiv preprint arXiv:1911.05722, 2019
>
> [3] Timothy P. Lillicrap et al, Continuous control with deep reinforcement learning, arXiv preprint arXiv:1509.02971, 2015
>
> [4] Chen, X. and He, K. Exploring simple siamese representation learning. arXiv preprint arXiv:2011.10566, 2020.
>
> [5] Jean-Bastien Grill et. al, Bootstrap your own latent: A new approach to self-supervised learning. arXiv:2006.07733v1, 2020.

---

### Official Review · Reviewer_GsyJ · 2021-11-06

**Correctness:** 4
**Technical Novelty And Significance:** 3
**Empirical Novelty And Significance:** 3
**Recommendation:** 8
**Confidence:** 5

**Main Review:**

# Strengths
+ The proposed method is easy to follow and the paper is well-organized.
+ The BGRL model is simple and effective.
+ Extensive experiments demonstrate the effectiveness and efficiency.

# Weaknesses
- The novelty of this work is a bit limited as this is a simple adaptation of BYOL on graph-structured data.


**Summary Of The Paper:**

This paper presents a novel graph contrastive learning model with bootstrapping latent objectives. Extensive experiments demonstrate the effectiveness of the proposed BGRL method.

**Summary Of The Review:**

It feels to me that no obvious weaknesses can be spotted. The BGRL method is easy to follow, is simple and also powerful, and the authors did an extensive set of experiments to verify the effectiveness and also efficiency of BGRL.

# Suggestions
* In Table 5, the memory consumption for both GRACE and BGRL should be given so that readers could understand how much memory is needed for sufficient number of negative samples to achieve similar performance as with BGRL.
* I believe some graph-level evaluations may further expand the applicability of BGRL. In this case, how to choose appropriate representations (e.g., graph- or node-level embeddings or some sort of context representations) for both branches is important.
* As the authors point out, theoretical analysis on the learning dynamics of the online and the offline branch [1] may be helpful for understanding the superior performance of BGRL.

[1] Y. Tian, X. Chen, and S. Ganguli, Understanding Self-Supervised Learning Dynamics without Contrastive Pairs, in ICML, 2021.

---

> ### Author Response · Authors · 2021-11-17
> **Response to reviewer GsyJ**
>
> Thank you very much for the suggestions and positive feedback. We are glad that you found the paper well-written and empirically strong.
>
> (1) Memory consumption comparisons:  Based upon your suggestion, we reran the experiments on ogbn-arXiv in Table 5 to include measurement of memory consumption of GRACE and report the results in the table below. In this experiment, we show that BGRL uses roughly half the memory of GRACE while achieving higher accuracy.
>
> | Method                     | Memory Used | Accuracy     |
> |----------------------------|-------------|--------------|
> | BGRL                       | **4.84 GiB**    | **71.64 ± 0.12** |
> | GRACE-SUBSAMPLING (k=2048) | 9.59 GiB    | 71.51 ± 0.11 |
> | GRACE-SUBSAMPLING (k=32)   | 5.68 GiB    | 71.18 ± 0.16 |
> | GRACE-SUBSAMPLING (k=8)    | 5.63 GiB    | 70.33 ± 0.18 |
> | GRACE-SUBSAMPLING (k=2)    | 5.62 GiB    | 60.24 ± 4.06 |
>
> _Memory requirements for GRACE and BGRL to accompany the results in Table 5._
>
>
> (2) Graph-level evaluations:  We agree that easily being extendable to other applications such as learning graph-level embeddings is a strength of this approach. As our focus was on scalability, we sought to dig deeper into node-based predictions on very large graphs, rather than applying BGRL to graph-level inference. Extending BGRL to graph-level prediction  is a fruitful direction for future work, particularly as the problem of choosing informative negative examples at the graph level is still unsolved.
>
> (3) Discussion of learning dynamics:  Thank you for your suggestion. We will add additional discussion of the results from [1] to give further insight into the learning dynamics of BGRL.
>
> [1] Y. Tian, X. Chen, and S. Ganguli, Understanding Self-Supervised Learning Dynamics without Contrastive Pairs, in ICML, 2021.

---

> > ### Comment · Reviewer_GsyJ · 2021-11-27
> > **Response to rebuttal**
> >
> > Thanks for your response. Please include these discussions in your final version. Again, I find the empirical results valuable to graph contrastive learning and thus I would love to see this work being accepted to ICLR.

---

> > > ### Author Response · Authors · 2021-11-29
> > > **Thank you for the response**
> > >
> > > Thanks for your continued positive view of our work. We are glad these points addressed your questions and will definitely add a discussion of this to the final version of the paper.

---

### Official Review · Reviewer_CbdK · 2021-11-08

**Correctness:** 4
**Technical Novelty And Significance:** 2
**Empirical Novelty And Significance:** 3
**Recommendation:** 6
**Confidence:** 5

**Main Review:**

Strengths:

The developed method is technically sound. The paper is well-organized and easy to follow. The motivation of employing BYOL in tackling the scalability challenge is clearly stated and the empirical studies are sufficient to show the efficacy of BGRL in both producing high quality node representations and scaling to extremely large graphs. I appreciate the empirical results provided by the paper, which is useful to the graph representation learning community. The implementation specifications are recorded in detail. Other contributions are summarized as above.

Weaknesses:

(1) My major concern is the novelty of the paper. Although dropping projection network makes differences in the model architecture between BGRL and the original BYOL, its technical contribution is incremental. Besides, such technique is already discussed in SelfGNN [1].

(2)  In experiments on large graphs, it would increase the significance if

* an ablation study on the model choice of encoder is provided,
* more recent base model(s) with O(n) complexity in computing objective functions (e.g., MVGRL [2]) are compared against.



References

[1] Zekarias T. Kefato and Sarunas Girdzijauskas. Self-supervised graph neural networks without explicit
negative sampling. CoRR, abs/2103.14958, 2021

[2] Kaveh Hassani and Amir Hosein Khasahmadi. Contrastive multi-view representation learning on
graphs. In International Conference on Machine Learning, 2020.

**Summary Of The Paper:**

The paper proposes a self-supervised graph representation learning algorithm named BGRL. BGRL employs similar architecture & training mechanism with BYOL and necessary adjustments (e.g., the augmentation methods, dropping the prejector module) to adapt to the characteristics of graph datasets. Given the nature of BYOL-based methods that negative samples are not used during training, BGRL scales only to O(n) when computing the objective function, which essentially breaks the computation limit incurred by the quadratically complex objective function adopted by SimCLR-based models (e.g., GRACE), hence makes the proposed model applicable to large graphs.

The major contributions of the paper are summarized as follows:

(1) The paper highlights and empirically verifies BYOL's advantages in not only boosting task performance but also enhancing model scalability, the second aspect is important but ignored by previous / concurrent BYOL-based methods.

(2) The paper tests the effect of introducing label supervision to the BYOL-based scheme on learning node representations for large graphs, which is not studied in previous / concurrent BYOL-based methods.

(3) The paper evaluated BGRL against other base models on various benchmarks, which not only provides multiple aspects of BGRL's advantages, but also enriches the baselines of unsupervised graph representation learning, especially on large graphs.

**Summary Of The Review:**

In summary, although there are empirical contributions in the proposed method, the paper appears only to apply the well-developed BYOL scheme to graph representation learning tasks. The technical contribution is sort of incremental.

---

> ### Author Response · Authors · 2021-11-17
> **Response to reviewer CbdK**
>
> Thank you for the valuable feedback and suggestions. We are glad you found the goal of addressing the scalability challenge in graph representation learning well-motivated and that BGRL effectively addresses this problem.
>
> (1) Novelty of the paper:  In response to your concern about the novelty, please see our General Response for all reviewers. We point out the key contributions of our work that go beyond the algorithmic innovations, and highlight our emphasis on scalability of BGRL over other methods in the graph domain.
> With regards to novelty relative to SelfGNN, while we cannot provide explicit reference to maintain anonymity, our work was posted to arXiv before SelfGNN (and presented in a workshop) and thus would be considered concurrent work. Further, we examine the application of bootstrapping to the graphs domain exhaustively and demonstrate scalability properties crucial for general applicability in realistic settings, while SelfGNN provides a more limited empirical analysis on small graphs.
>
>
> (2) Further architecture ablations for large-scale experiments:  We opted for MPNNs for the MAG240M dataset, (i) due to their higher performance on this complicated task over simpler models, and (ii) to gauge the effectiveness of training these more complex models through unsupervised methods.
> Following your suggestions, we run additional experiments with simpler GCN encoders instead and report our results in Appendix H, Figure 10. We see that BGRL is able to improve performance over the fully-supervised approach even with these weaker encoders, and also learn faster and more stably than GRACE.
>
>
> (3) Comparisons with MVGRL:   The MVGRL [1] preprocessing step computes a graph diffusion as an augmentation, using methods such as Personalized PageRank, Heat Kernel, or Floyd-Warshall all-pairs distance computations to achieve good performance, each of which requires O(N^3) time to compute. Such augmentation methods simply cannot scale to the large graphs of hundreds millions of nodes that we consider. Further, the graph diffusion they output may be dense and add edges not previously present in the graph - thus although the update step is linear in the size of the augmented graph, it may be quadratic in the size of the original input graph.
>
> In contrast, BGRL relies only on relatively simple and memory-efficient augmentations. Further, we show in Table 4 that the performance of BGRL using only these simple augmentations is competitive with BGRL using more complex augmentations based on PPR or eigenvector methods, thus indicating that BGRL can safely rely on simple augmentations to scale to realistic graphs without sacrificing performance.
>
> Finally, when training from a batch of subsampled neighborhoods, MVGRL’s update step is actually quadratic in the batch size (as can be seen clearly from Algorithm 1 in [1]), whereas BGRL is linear, allowing us to more easily scale up to larger batch sizes.
>
>
> [1] Kaveh Hassani and Amir Hosein Khasahmadi. Contrastive multi-view representation learning on graphs. In International Conference on Machine Learning, 2020

---

> > ### Comment · Reviewer_CbdK · 2021-11-27
> > **Thanks for rebuttal**
> >
> > Thanks for the response from authors, that clears my questions on the experiments. I agree with the point made by authors in the general response that the contribution of the paper is from empirical studies, and I find the results provided in the paper valuable to the community. Therefore I would recommend the paper to be accepted, I have adjusted my recommendation score accordingly.

---

> > > ### Author Response · Authors · 2021-11-29
> > > **Thank you for the update**
> > >
> > > Thank you for your positive response and for updating the score.
> > > We are glad these points addressed your concerns and we will be sure to add them to the final version of the paper.

---

### Author Response · Authors · 2021-11-17
**General Response to Reviewers**

We thank the reviewers for their feedback and constructive suggestions. We are pleased that many reviewers appreciated the contributions of our work, said the “method is technically sound” (CbdK), “well-organized and easy to follow” (CbdK, GsyJ), “experimental contribution is very solid” (8wKC), and generally agreed that our study of the memory-performance scalability tradeoff in graph representation learning was insightful.

At the same time, multiple reviewers expressed some concerns about the novelty of the approach. While it is true that the loss and architecture underlying BGRL are an “adaptation of BYOL on graph-structured data" (GsyJ), we want to emphasize that applying algorithms in new domains is not trivial, and that understanding how approaches transfer to new domains requires evaluations to characterize their performance along a number of different axes.

An important contribution of our work is to analyze the computational efficiency vs. performance tradeoff in contrastive methods, and to show that by eliminating negative examples from latent predictive approaches, BGRL can achieve the same or better performance at a fraction of the memory and computation of other SOTA methods such as GRACE. Combining the memory efficiency of BGRL with neighborhood-sample based updates allows us to scale to extremely large graphs. We also provide experiments in the semi-supervised setting on massive real-world datasets which have not been demonstrated by prior works, further highlighting the flexibility and scalability of our approach.

While algorithmic and architectural innovations are critical for driving progress in the field, robust empirical investigations and studies of different aspects of performance (i.e., memory and scalability in our case) are also essential for advancing our collective understanding of how nascent approaches like BYOL work in new domains.^ We believe that the robust and extensive analysis that we have provided for different graph representation learning methods, across a variety of tasks, augmentations, and ablations, is an extremely important resource to the community. Further, we have already prepared for open-sourcing our code to reproduce our results, thus making it more useful in practice.


^ Note: We would like to highlight the explicit emphasis on empirical contributions in ICLR this year. In a call for papers, the program chairs write that “Submissions with significant contributions in either technical aspects or empirical aspects will be given high priority for acceptance”.

---

### Author Response · Authors · 2021-11-25
**Update on Rebuttal**

We thank the reviewers again for their thoughtful reviews and comments.

We believe we have addressed all of the reviewers' concerns in our rebuttal, and are happy to continue the discussion further before the rebuttal period ends.

Please let us know if you have any further questions or if your concerns have been addressed.

---

### Decision · Program_Chairs · 2022-01-20

**Decision:**

Accept (Poster)

**Comment:**

To perform self-supervised graph representation learning that is scalable to large graphs, the authors propose Bootstrapped Graph Latents (BGRL) that learns its graph representation by predicting alternative augmentations of the input, avoiding the need to construct negative examples. The weakness of the paper lies in its novelty, as it can be considered as a direct adaptation of the BYOL method, whose success has been demonstrated on self-supervised visual representation learning, to learn graph node representations. While the novelty is limited, the paper has shown how to appropriately apply BYOL to graph representation learning, achieving state-of-the-art results on graph node representation learning on large-scale graphs. The overall assessment of the reviewers is that the empirical significance of the paper outweighs its shortcoming in novelty. The AC agrees with this assessment and hence recommends acceptance.